# The Common Good Balance Sheet and Employees' Perceptions, Attitudes and Behaviors

**Jasmin Wiefek \* and Kathrin Heinitz**

Arbeitsbereich Arbeits- und Organisationspsychologie, Freie Universität Berlin, Habelschwerdter Allee 45, 14195 Berlin, Germany; kathrin.heinitz@fu-berlin.de
\*   Correspondence: jasminwiefek@zedat.fu-berlin.de

**Abstract:** The Common Good Balance Sheet (CGB) is an instrument to measure a company's contribution to the common good. In our study, we investigate whether employees from companies with higher CBG scores perceive more corporate social responsibility than employees from companies with lower CBG scores and whether relationships can be found between the achieved CGB scores and employees' job-related attitudes and behaviors. We conducted an online survey of 332 employees from eight German companies with published CGBs. According to results from multiple linear regression analyses, employees from companies with higher CGB scores perceive more CSR and are more satisfied with their jobs and payments. In addition, they report less job demands, more organizational support, more work meaningfulness and more organizational citizenship behaviors towards their company. Employees identify more with their company if high transparency and co-determination is practiced. However, the value and social impact of the companies' products is not related to employees' organizational identification. Moreover, employees from companies with high CGB scores do not report more organizational citizenship behaviors towards their colleagues. Our results indicate that the CGB is a tool that measures aspects concerning job-related attitudes and behaviors and allows comparability between companies. However, aspects relevant to job satisfaction may still be missing in the CGB scoring.

**Keywords:** micro-CSR; economy for the common good; job satisfaction; pay level satisfaction; job demands; work meaningfulness; organizational identification; perceived organizational support; organizational citizenship behavior; SMEs

## 1. Introduction

In 2015, the United Nations (UN) summit in New York adopted the 2030 Agenda for Sustainable Development. With the 2030 Agenda, the UN set itself 17 Sustainable Development Goals (SDGs) in order to tackle globally poverty, inequality, injustice and climate change. Goal number eight consists of the requirement to, "make sure that financial progress creates decent and fulfilling jobs while not harming the environment" [1]. Companies are therefore also required to contribute to achieving the SDGs. Companies can use corporate social responsibility (CSR) management tools such as the SDG Compass, the EcoManagement and Audit Scheme (EMAS), the Global Reporting Initiative (GRI) or the International Organization for Standardization (ISO) norms in order to document and communicate their social and ecological commitment.

For a long time, the focus in CSR research was on large companies, which is also why many of the CSR theories were developed with large companies in mind [2]. The development of many of the CSR management tools was also therefore focused on large companies, although it was assumed that the tools could be downscaled so they would also be suitable for small and medium-sized enterprises (SMEs) [2]. However, the idea of downscaling is based on particular assumptions that do not necessarily apply to the

SMEs [2]. In practice, SMEs have had difficulty adapting CSR management tools such as EMAS, ISO 14001 and GRI to their internal processes, as well as problems with managing the required documentation of such standards [3–5]. In addition, the modified versions of the CSR standards adapted to SMEs are hardly used in practice [3,6].

In contrast to the established CSR management tools, the Common Good Balance Sheet (CGB) developed by the Economy for the Common Good (ECG) is a tool that works primarily for small and medium-sized companies [7]. With the aid of the CGB, the SMEs can systematically record their otherwise implicit social and ecological commitment and communicate this both internally and externally. The ECG regards the contribution to the common good, operationalized through socio-ecological practices, as the primary purpose of all business activity. The aim of the ECG is to use the CGB to translate companies' common good commitment into comparable figures in order, for example, to award public contracts or credits depending on performance in the CGB. The aim is to ensure the comparability of the results in the CGBs through peer review processes or external auditing [8].

But what do the figures from the CGBs really say about everyday working life in the companies? To what extent does what is documented formally, and mostly at management level, extend into the workforce? Our study aims to explore these aspects. We therefore ask the following questions: (a) do employees from ECG companies perceive their companies' common good orientation?; and (b) do companies' CGB scores correlate with employees' attitudes, such as work and pay level satisfaction, work demands, perceived organizational support, organizational identification, meaningfulness of work, and employees' organizational citizenship behaviors?

*The Common Good Balance Sheet*

The social movement of the ECG advocates an economic system based on values that promote the common good. Its aim is to be a catalyst for change on the economic, political and social level [9]. It was founded by a small group of entrepreneurs together with the activist and writer Christian Felber in Austria in 2010 [10]. In 2015, the European Economic and Social Committee (EESC) recommended the ECG model be integrated into the legal framework at both the European and national level [11]. In Germany, some state governments then took the decision to implement a CGB within individual state-owned companies or to support companies in compiling a CGB, e.g., [12,13]. In 2019, a motion was proposed at the federal level for a pilot project in which a CGB would be implemented in at least two companies wholly or partially owned by the German federal government [14].

According to ECG data, 2000 companies support the ECG and around 400 companies are either a member or have already compiled a CGB (as of March 2020, [15]). Common good-oriented companies (CgoCs) operate within the framework of "profit satisficing", which means their objective is not profit maximization, which leaves scope for pursuing socio-ecological principles [7]. In the CGB, the contribution to the common good is measured using the value groups of human dignity, solidarity and cooperation, ecological sustainability, social justice, democratic co-determination and transparency. In their documentation, the companies make reference to their suppliers, their investors, their employees, their customers, and businesses in the same field, as well as to the environment and the social environment (CGB version 4.1, see Table 1). Version 5.0 of the CGB is now available. In our study, however, we will work with version 4.1 because the companies we have investigated have compiled their CGBs using this version.

**Table 1.** Common Good Balance Sheet (Matrix 4.1). Source: Handbuch zur Gemeinwohl-Bilanz [16] (part I).

| | Human Dignity | Cooperation & Solidarity | Ecological Sustainability | Social Justice | Co-determination & Transparency |
|---|---|---|---|---|---|
| **(A) Suppliers** | **A1: Ethical supply management:** Active examination of the risks of purchased goods and services, consideration of the social and ecological aspects of suppliers and service partners (90) | | | | |
| **(B) Investors** | **B1: Ethical financial management:** Consideration of social and ecological aspects when choosing financial services; common good-oriented investments and financing (30) | | | | |
| **(C) Employees, including Business Owners** | **C1: Workplace quality and affirmative action:** Employee-oriented organizational culture and structure, fair employment and payment policies, workplace health and safety, work-life balance, flexible work hours, equal opportunity and diversity (90) | **C2: Just distribution of labor:** Reduction of overtime, eliminating unpaid overtime, reduction of total work hours, contribution to the reduction of unemployment (50) | **C3: Promotion of environmentally friendly behavior of employees:** Active promotion of sustainable lifestyle of employees (mobility, nutrition), training and awareness-raising activities, sustainable organizational culture (30) | **C4: Just income distribution:** Low income disparity within a company, compliance with minimum and maximum wages (60) | **C5: Corporate democracy and transparency:** Comprehensive transparency within the company, election of managers by employees, democratic decision making on fundamental strategic issues, transfer of property to employees (90) |
| **(D) Customers, Products, Services, Business Partners** | **D1: Ethical customer relations:** Ethical business relations with customers, customer orientation and co-determination, joint product development, high quality of service, high product transparency (50) | **D2: Cooperation with businesses in same field:** Transfer of know-how, personnel, contracts and interest-free loans to other business in the same field, participation in cooperative marketing activities and crisis management (70) | **D3: Ecological design of products and services:** Offering of ecologically superior products/services; awareness raising programmes, consideration of ecological aspects when choosing customer target groups (90) | **D4: Socially oriented design of product and services:** Information, products and services for disadvantaged groups, support for value-oriented market structures (30) | **D5: Raising social and ecological standards:** Exemplary business behavior, development of higher standards with businesses in the same field, lobbying (30) |
| **(E) Social Environment** | **E1: Value and social impact of products and services:** Products and services fulfill basic human needs or serve humankind society or the environment (90) | **E2: Contribution to the local community:** Mutual support and cooperation through financial resources, services, products. logistics, time, know-how, knowledge, contracts, influence (40) | **E3: Reduction of environmental impact:** Reduction of environmental effects towards a sustainable level, resource, energy, climate, emissions, waste etc. (70) | **E4: Investing profits for the common good:** Reduction or eliminating dividend payments to extern, payouts to employees, increasing equity, social-ecological investments (60) | **E5: Social transparency and co-determination:** Common good and sustainability reports, participation in decision-making by local stakeholders and NGO's (30) |
| **Negative Criteria** | Violation of ILO norms / human rights (–200), products detrimental to human dignity and human rights (e.g. landmines, nuclear power, GMO's) (–200), outsourcing to or cooperation with companies which violate human dignity (–150) | Hostile takeover (–200), blocking patents (–100), dumping prices (–200) | Massive environmental pollution (–200), gross violation of environmental standards (–200), planned obsolescence (short lifespan of products) (–100) | Unequal pay for women and men (–200), job cuts or moving jobs overseas despite having made a profit (–150), subsidiaries in tax havens (–200), equity yield rate >10% (–200) | Non-disclosure of subsidiaries (–100), prohibition of a work council (–150), non-disclosure of payments to lobbyists (–200), excessive income inequality within a business (–150) |

The CGB consists of 17 indicators. Each indicator formulates particular aims and requirements of the ECG, asks thought-provoking questions and describes what the implementation of these requirements within companies might look like. The extent to which a company is fulfilling the requirements is captured by a percentage or score. The scores from the individual indicators are added together to produce an overall score. In the CGB 4.1, it is possible to achieve values between −2350 and 1000 points if negative criteria are included [16].

In a factor-analytical study, the measurement model on which the CGB 5.0 is based did not prove to be valid and reliable [17,18]. Ejarque and Campos [17] identified items that need to be removed in order for the model to hold. However, further research is still needed to redefine them and retest the measurement model with the redefined items [17]. Therefore, the development of the CGB is an ongoing process.

## 2. Review of the Literature

In the first few years of the ECG, the publications on the ECG were mainly bachelor's and master's theses and shorter informal studies. Since around 2016, longer publications and articles have appeared in scientific journals [19]. Kny [19] concludes from a synopsis of the literature that, in comparison with prevalent CSR approaches, the standards the ECG sets with a view to socio-ecological change are thematically and normatively extensive. The ECG's work-related values can be summarized to avoid discrimination and boost quality of employment, to encourage the sharing of information and worker participation, and to promote beneficial psycho-social factors at work, including flexible working hours, work-life balance, task clarity and variety of tasks, scope and autonomy [20].

Ollé-Espluga et al. [20] conducted a descriptive analysis of reported work and employment conditions in 59 CGBs from German and Austrian companies and compared their results to the quality of jobs in the Austrian and German economies overall. According to their results, CgoCs provide more favorable conditions in terms of training and control over daily working time and tasks. Furthermore, the possibility of working part-time and at home is more prevalent in CgoCs, and more CgoCs report the existence of direct participation practices instead of representative participation forms. However, when compared to the Austrian and German economies overall, CgoCs stand out for their higher prevalence of works councils. Moreover, CgoCs report limited use of precarious employment arrangements, and almost half of CgoCs provide full-time salaries higher than or equal to the reference value for living wages in Austria and Germany. Comparison data with the German and Austrian economies overall was not available at the time of the study. Contrary to expectations, Ollé-Espluga and colleagues [20] observed a slightly greater wage inequality between the highest and the lowest income in CgoCs than in the Austrian and German economies overall.

In an explorative interview study, Meynhardt and Fröhlich [21] investigated how a CGB can contribute to the development of companies from the perspective of the companies themselves. The company representatives reported, for example, an increased awareness around interacting with employees, awareness creation around reducing workload, women in leadership roles, workplaces with disabled access, and diversity of opinion, as well as the introduction of behavioral codes and a re-examination of ownership structures. In addition, the CGB increased transparency for both internal and external stakeholders [21].

Sanchis et al. [22] conducted a quantitative questionnaire study among 206 European companies in order to assess the impact of the ECG model at the organizational level. Some companies reported, for example, improvements in cooperation strategies among businesses and better relations with suppliers, improved participation and better communication with employees and leadership as well as improvements in employees' commitment and better levels of employee motivation and satisfaction since conducting their first CGB. However, the study was not able to establish conclusively whether all

these improvements were attributable to the production of the CGBs [22]. This may be due to the fact that most of the companies—as we also found out from our own interviews with CgoCs—were already socio-ecologically committed before the CGB, so the CGB served, to a certain extent, as a tool for documentation and communication rather than as a driver of change [23].

In terms of the effect on employees, we may assume, based on the results of a qualitative interview study by Mischkowski et al. [24], that the CGB is having a positive impact on employee retention. Here, Mischkowski et al. [24] emphasize the aspects of participation and co-determination within companies, as well as the establishment of a clear value basis, which also creates a point of orientation for interactions both within the company and with external stakeholders. Thus far, there have been no other studies on the effect of the common good commitment on employees within CgoCs. Within CSR research, however, there have already been studies on the effects of socio-ecological commitment on employees.

### 2.1. Employees' Reactions to CSR

According to Glavas and Kelley [25] definition of CSR, CSR means a company caring for the well-being of its employees and other key stakeholders, including the social and natural environment, with the aim of also creating value for the business. CSR is manifested in corporate strategies and operating practices [25]. Therefore, we understand a company's common good contribution, as the ECG defines it, as a form of CSR. Perceived CSR refers to the extent to which employees perceive the development and implementation of CSR strategies and practices within their company as well as the CSR′s impact on the well-being of all key stakeholders and the natural environment [26,27].

For a long time, there was very little focus on the employee level within scientific CSR research, but over the past few years the number of micro-CSR studies has increased significantly [28,29]. Here, micro-CSR means, "the study of the effects and experiences of CSR (however it is defined) on individuals (in any stakeholder group) as examined at the individual level" [30] (p. 216). According to a review of 268 articles by Gond et al. [29], most of the studies on micro-CSR published thus far can be divided into three streams of research: (a) drivers of CSR engagement, which relates to the predictors of, motives for, or forces that trigger employees' CSR engagement; (b) evaluations of CSR, which means how employees perceive, experience and judge their employers' CSR practices; and (c) reactions to CSR, which concerns the individual-level reactions to CSR and the underlying mechanisms and individual-level boundary conditions involved.

In a meta-analysis of 65 studies from 67 samples, Wang et al. [31] have studied the reactions to perceived CSR from a summative perspective. They come to the conclusion that, "perceived CSR is positively correlated with employees' positive attitudes and behaviors, and negatively correlated with employees' negative attitudes and behaviors" [31] (p. 18). The results indicate, for example, that perceived CSR is positively correlated with employees' positive beliefs and attitudes, such as perceived external prestige (rc = correlations controlling measurement and sampling error: rc = 0.378), perceived organizational support (rc = 0.699), organizational identification (rc = 0.515), organizational trust (rc = 0.532), organizational commitment (rc = 0.538), organizational justice (rc = 0.551), work engagement (rc = 0.515) and job satisfaction (rc = 0.520). In addition, perceived CSR is positively correlated with employees' positive behaviors, such as job performance (rc = 0.483) and organizational citizenship behavior (rc = 0.405) [31].

If we distinguish, for example, between internal CSR (CSR directed at employees) and external CSR (CSR directed at external stakeholders), Wang et al. [31] report that perceived internal CSR correlated significantly and positively with employees' organizational identification (rc = 0.575) and work engagement (rc = 0.787), but that the correlation with job satisfaction was not significant (rc = 0.264, ns.). With respect to perceived external CSR, they found positive correlations with employees' organizational identification (rc = 0.489), work engagement (rc = 0.727) and job satisfaction (rc = 0.589) [31].

Moreover, the meta-analysis showed that perceived CSR towards the public and environment is positively correlated with employees' organizational trust (rc = 0.272), job satisfaction (rc = 0.427) and organizational citizenship behavior (rc = 0.410) [31]. However, the relationships between organizational identification and perceived CSR towards employees (rc = 0.421, ns.) and the environment (rc = 0.318, ns.) were not significant [31].

The Role of Sex and Age in Reactions to CSR

Wang and colleagues [31] have also highlighted the role of sex and age in relation to the reactions to CSR. Although they assume, on the basis of previous studies, that the impact of perceived CSR on employees' attitudes and behaviors tend to be more evident among females [32,33], the results of their study contradict this hypothesis. When the proportion of males in the sample increased, the relationship between perceived CSR and attitudinal variables such as external prestige ($\beta = -1.136$, $p < 0.001$) and work engagement ($\beta = -1.441$, $p < 0.05$) was weakened, whereas the relationship between perceived CSR and behavioral variables such as employees' job performance ($\beta = 0.807$, $p < 0.05$) and organizational citizenship behavior ($\beta = 0.416$, $p < 0.001$) was strengthened [31]. Also, Islam et al. [34] and Ko et al. [35] found the relationship between employees' perceptions of CSR and organizational identification to be stronger among men than women. Hence, there is evidence of the role of sex in relation to reactions to CSR, though the exact direction of correlations still needs to be clarified.

With respect to age, Wang et al. [31] found the relationships between perceived CSR and organizational trust ($\beta = 0.037$, $p < 0.05$), job satisfaction ($\beta = 0.024$, $p < 0.01$) and organizational deviance ($\beta = 0.060$, $p < 0.01$) to be more significant among older employees, while the relationships between perceived CSR and employees' work engagement ($\beta = -0.038$, $p < 0.01$), job performance ($\beta = -0.025$, $p < 0.05$) and creativity ($\beta = -0.058$, $p < 0.001$) were more significant among younger employees. The moderating effect of average age on the relationship between perceived CSR and employees' organizational identification and organizational commitment was not significant [31]. Thus, influences of age on reactions to CSR should be taken into account when conducting micro-CSR studies.

*2.2. Research Desiderata and the Aim of Our Study*

Thus far, there has been relatively little research on theories and empirical studies of CSR and work meaningfulness [28]. Rosso et al. [36] assume that companies' emphasis on their contribution to the common good may have positive implications for employees' experience of meaningfulness. Furthermore, Aguinis and Glavas [37] point out that CSR could be used to create corporate cultures that are caring and compassionate. They join other scholars in calling for more research on caring and compassionate organizational cultures in order to shift away from the predominant focus on management in cultures marked by aggressiveness, competitiveness and rigid norms. Due to the relational nature of CSR, future research ought to explore how creating caring relationships, such as caring for the well-being of stakeholders, has an impact on employees [28]. At the same time, Glavas [28] observes that there are still too few studies on micro-CSR in SMEs.

Thus far, only SMEs have published a CGB, which means our study on micro-CSR in CgoCs is helping to fill a research gap. The aim of the ECG is to establish an economy based on cooperation and solidarity. Accordingly, these values must be reflected not only in business dealings with external stakeholders but also within the company cultures themselves. This is another respect in which our study on the ECG fills a research gap. Studies conducted thus far on the effects of a common good orientation (CGO) at the organizational level are of an exploratory nature and merely capture the opinions and perspectives of individual people from the respective companies. Often, the individuals consulted for the purpose of the studies are those who are involved in compiling the CGB. Interviews with 11 CgoCs for one of our earlier studies show that these individuals are frequently directors or employees with a managerial role [38]. The aim of our study is to clarify, for the first time, if employees perceive the CGO of their companies. In addi-

tion, we aim to establish whether there is any relationship between the companies' CGO and employees' work-related attitudes and organizational citizenship behaviors. In so doing, we also look at aspects of work meaningfulness.

## 3. Constructs and Hypotheses

Below, we introduce the constructs used in our study and the hypotheses tested.

### 3.1. Perceived CSR

The scores from the CGBs serve as comparable indicators of the extent of CGO within a company. In general, CSR measures can only have an influence on the attitudes and behavior of employees to the extent these employees perceive and evaluate the CSR engagement [39]. The CgoCs either implement their CGO top-down via a delegation system or take a less institutionalized approach, allowing CGO to be implemented bottom-up by the collective; this is primarily the case with collectively owned companies [7]. If CGO is anchored within the company, the CSR commitment connected with it is also part of the company's daily operations, and every employee should come into contact with this in some capacity [40]. Because a higher score in the CGB is supposed to be an indicator of greater CGO, we may assume that an increasing score correlates with an increase in perceived CSR. We therefore propose the following hypotheses.

**Hypothesis 1 (H1).** *The total score achieved in the CGBs is positively related to perceived CSR.*

### 3.2. Job Satisfaction

A widely used definition of job satisfaction is one proposed by Locke [41] (p. 1304), "a pleasurable or positive emotional state resulting from the appraisal of one's job or job experiences". Ollé-Espluga et al. [20] (p. 4) attest to, "a widespread presence of elements of good quality of work" within CgoCs. According to relationship management theory, CSR practices are an effective tool in improving the relationship between companies and their employees [30,42]. Bauman and Skitka (2012) explain how CSR may provide employees with a sense of security with regard to their material needs being met, self-esteem that stems from a positive social identity, as well as feelings of belongingness and meaningfulness at work, all of which may improve employees' job satisfaction. This is why we believe that job satisfaction is higher in companies with a higher score in the CGB. We therefore propose the following hypothesis.

**Hypothesis 2 (H2).** *The total score achieved in the CGBs is positively related to overall job satisfaction.*

### 3.3. Job Demands and Perceived Organisational Support

Job demand is measured in the CGB with the indicator C2: just distribution of labour. The relevant figures are the proportion of all-inclusive work contracts, the overtime worked per employee, the proportion of part-time employees in the company, the number of new appointments, and the number of employee surveys on working hours and working time models. The background to the indicator is the ECG's ambition for a "just" distribution of workload among all people capable of employment, as well as a reduction in regular weekly working hours [16].

The indicator C1: workplace quality and affirmative action encourages companies to investigate and reflect on employee-oriented organizational cultures and structures, the promotion of health and safety in the workplace, work-life balance and flexible working hours, as well as equal opportunities and diversity. The relevant figures are inter alia, the take-up of workplace offerings related to physical and mental healthcare, as well as the number of occupational accidents, employees on long-term sick leave and employees who have taken early retirement as a result of inability to work [16].

The aims of the two indicators are to keep the workplace demands on the employees as low as possible. Hence, we may assume that work-related demands should be reduced if a company scores highly in indicators C2 and C1. We therefore propose the following hypotheses.

**Hypothesis 3 (H3).** *The scores achieved in indicator C2: just distribution of labor are negatively related to job demands.*

**Hypothesis 4 (H4).** *The scores achieved in indicator C1: workplace quality and affirmative action are negatively related to job demands.*

The extent to which employees perceive that their companies value the employees' contributions and pay attention to their well-being is defined as, "perceived organizational support (POS)" [43]. Thus, POS should be shaped by the way a company treats the employees [44]. POS has been found to be positively related to CSR [25]. Similarly, a positive correlation between perceived CSR and POS was found in the meta-analysis by Wang et al. [31]. Thus, we hypothesize that POS should improve with increasing scores in indicator C1.

**Hypothesis 5 (H5).** *The scores achieved in indicator C1: workplace quality and affirmative action are positively related to perceived organizational support.*

*3.4. Pay Level Satisfaction*

The ECG's aim is to ensure a "just" and transparent distribution of pay and profits within companies based on the standards it sets out. The aim of the indicator C4: just income distribution" is to measure the income distribution across a company. The compensation should be based on the employee's performance, the labor and responsibility involved in the role, the risks associated with the workplace and the necessity of the role. The ECG enquires into the lowest and highest wages within the company, the median income, and whether the company's internal compensation system is transparent. According to ECG standards, an internal income distribution of maximum 1:4 is the ideal. Companies with a distribution of 1:12 are heading in the right direction. The minimum income should adequately meet the living costs of the respective country and region in which the company is engaging the employees [16].

The aim of the indicator E4:investing profits for the common good is to measure the extent to which the profits made by a company are distributed or reinvested as fairly and meaningfully as possible, as well as in ways that promote the common good [16]. According to the ECG, incomes should in principle be connected with performance, and capital ownership should not represent any claim to an income. The ECG makes an exception in the case of a "company founder pension," which the founders of a company could receive for a period of time equal to the time they had spent actively building up the company [16].

The extent to which employees are, or are not, satisfied with their pay is described as pay satisfaction. Pay satisfaction encompasses the "amount of overall positive or negative affect (or feelings) that individuals have toward their pay" [45] (p. 246). Pay is understood here as all forms of remuneration, including direct cash payments such as salary, but also indirect non-cash payments such as benefits. The construct of pay satisfaction also includes the amount of pay rises and the process by which the compensation system is administered [46]. According to Williams et al. [46], different authors have suggested replacing this broad definition of pay satisfaction with a multidimensional conceptualization of pay satisfaction. One dimension of pay satisfaction is pay level satisfaction, defined as, "an individual's satisfaction with his or her base pay" [45] (p. 245).

According to the models of pay satisfaction, the pay level should have a direct influence on pay satisfaction and work satisfaction [47]. However, in a meta-study with 115

correlations from 92 independent samples, Judge et al. [47] found that pay level only modestly correlated with job satisfaction ($\varrho = 0.15$) and pay satisfaction ($\varrho = 0.23$). Hence, the absolute pay level has only, "little potential to satisfy" [47] (p. 164). In Lawler's discrepancy model of pay satisfaction, it should be the case that employees are satisfied with their pay when their perception of the pay received is equal to the amount they perceive they should be receiving [48]. If it is the employees' perception that they are receiving less pay than they believe they are entitled to, they become dissatisfied. On the other hand, if they receive more pay than they believe is appropriate, they may develop feelings of guilt, inequity and discomfort [48]. Hence, satisfaction with payment is primarily determined by perception and the fulfilment of expectations. Processes of social comparison in relation to the perceived pay of referent others also have a role to play in the model. The perception of one's own pay is influenced by what others are being paid and what others are being paid in relation to their input. Lawler assumes that the more one perceives what others receive, the less one perceives what one receives oneself and the more dissatisfied one becomes with one's own salary. Also, the greater the salary one perceives others to receive, the greater the expectation will be of what one should receive oneself [48].

The ECG's goal is a "just" and transparent distribution of income and profits. If we assume that the ECG's concepts of fairness correspond to those of the employees, pay level satisfaction in companies with higher values in indicators C4 and E4 should increase. We therefore propose the following hypotheses.

**Hypothesis 6 (H6).** *The scores achieved in indicator C4: just income distribution are positively related to pay level satisfaction.*

**Hypothesis 7 (H7).** *The scores achieved in indicator E4: investing profits for the common good are positively related to pay level satisfaction.*

*3.5. Meaningful Work*

In indicator E1:value and social impact of products and services, the ECG describes one of its goals as ensuring that global production does not exceed the level of what people really need for a sufficiency lifestyle, and, at the same time, ensuring that the production and supply of products and services are as socially-oriented and ecological as possible. In addition, the companies' range of products and services should contribute to poverty reduction and the resolution of social problems, as well as to food equality, education and health. The meaningfulness of products and services is measured by whether they satisfy a basic need and whether their production, use or disposal has negative consequences. Social impact is evaluated in terms of the personal growth of individuals, the strength of communities, and the sustainability of the natural environment [16].

The construct of meaningful work measures the extent to which employees perceive their work as significant. Meaningful work refers to the significance or value of work, which by definition has positive valence [49]. It can be understood as a unidimensional concept that captures a global judgement about whether one's work is perceived as worthwhile, important or valuable. Other scholars understand work meaningfulness as a multidimensional concept encompassing self-oriented concepts (such as self-actualization and personal growth) along with other-oriented concepts (such as helping others and contributing to the greater good) as an aggregate of meaningful experiences [50]. According to Allan et al. [50] (p. 501), experiences are meaningful, "when people conduct actions that fulfil values that are relevant to their existence and explain why their work is worth doing". If a company performs CSR, it sends signals to its employees that in addition to making a living, they are also serving others and society. This gives employees a sense that they are contributing to the common good and in turn helps employees find meaningfulness in their work [25]. Glavas and Kelley [25] found that employees' sense of meaningfulness is increased by perceived CSR only when actions are

directed towards third parties and not in terms of how the organization treats the employee.

On the whole, we may assume that increasing scores in indicator E1 correlate with an increasing perception of meaningful work. We therefore propose the following hypothesis.

**Hypothesis 8 (H8).** *The scores achieved in indicator E1: value and social impact of products and services are positively related to meaningful work.*

### 3.6. Organisational Identification

Organizational identification is a specific form of social identification [51] and reflects, "the extent to which individuals define the self in terms of the membership in the organization" [52] (p. 572). Thus, organizational identification is, "a perceived oneness with an organization and the experience of the organization's successes and failures as one's own" [53] (p. 103). Research suggests that CSR is positively related to organizational identification [26,34,35,54,55]. According to Wang et al.'s [1] meta-study, internal and external CSR as well as perceived CSR are in general positively correlated to organizational identification. However, the relationships between organizational identification and perceived CSR towards employees were not significant [31]. A study conducted by John et al. [55] exploring underlying processes suggests that if employees perceive organizational CSR positively, it will boost their pride in the company, which in turn affects the employees' organizational identification through the self-categorization process. According to the self-categorization theory, employees integrate into the companies that are most compatible with their values, with the aim of fulfilling their psychological desires for a meaningful existence and a sense of belonging [55]. Wang et al. [31], also describe, with reference to studies on potential congruences of values, how perceived CSR will promote employees' organizational identification based on signaling theory. The theory states thatorganizations signal with CSR the possibility of value fit between the organization and employees, through which employees may enhance their organizational identification [31].

Indeed, CgoCs report that by publishing CGBs, they find employees who are a better fit and share their values [24]. We may therefore assume that the employees in the CgoCs evaluate the CGO of their companies positively and propose the following hypothesis.

**Hypothesis 9 (H9).** *The scores achieved in indicator E1: value and social impact of products and services are positively related to organizational identification.*

The indicator C5: corporate democracy and transparency describes the ECG's ambition for employees to be involved in all essential decision-making (at least in their own area of operation) and executive personnel to be voted in and legitimized by employees. The ECG sees comprehensive transparency within the company as the prerequisite for this. The aim of the thought-provoking questions within this indicator is to establish whether all employees have access to critical information within the company, whether decision-making processes are democratic, what percentage of employees are involved in decision-making, and how transparent the decision-making processes are. Furthermore, companies are evaluated more highly if the employees are co-owners [16]. The ECG's aim here is to encourage extensive participation by employees in their companies. Wang et al. [31] refer to organizational identity theory and studies from micro-csr research when claiming that employees identify with organizations which meet their needs for sense of belonging, self-esteem, and self-identity through CSR activities, since employees are more likely to identify with organizations that may help them gain self-esteem and self-respect. Hence, we may assume that organizational identification positively corre-

lates with increasing scores in indicator C5. We therefore propose the following hypothesis.

**Hypothesis 10 (H10).** *The scores achieved in indicator C5: corporate democracy and transparency are positively related to organizational identification.*

*3.7. Organisational Citizenship Behaviours*

The ECG sees solidarity as one of its most fundamental values, demanding that companies demonstrate cooperation with other companies in indicator D2: cooperation with businesses in the same field. The ECG hopes that this will generate collaborations between companies, as collaboration—according to the ECG—fosters greater creativity, engenders new possibilities and more opportunities in the market and promotes better crisis absorption than when companies are in competition with one another. The companies are evaluated according to the extent to which they work together with other companies, mutually support one another (including financially), and make knowledge as well as financial and technical information available to one another. It is suggested, for example, that companies should exchange employees depending on the order situation [16].

The extent to which employees demonstrate solidarity and cooperative behavior is part of the construct called organizational citizenship behavior (OCB). OCB is an individual and initiative-taking behavior that is not part of the formal job requirements and serves to facilitate organizational functioning [56,57]. OCB-O is organizational citizenship behavior directed at the organization (e.g., attending functions that are not compulsory, and OCB-I is organizational citizenship behavior directed at individuals such as helping co-workers). Research indicates that CSR is positively related to OCBs [28,31,54,55]. Glavas [28] concludes from these results that if a company goes above and beyond its primary tasks (i.e., financial goals) and aims to contribute to the greater good of society by conducting CSR practices, then employees will go above and beyond their primary tasks and contribute to the greater good of the organization, demonstrating OCBs. Hence, we may assume that employees in companies with higher scores in indicator D2 report higher levels of OCB-O.

**Hypothesis 11 (H11).** *The scores achieved in indicator D2: cooperation with businesses in the same field are positively related to organizational citizenship behaviors directed at the company.*

CSR is also positively related to high-quality relationships among co-workers [58] and trust in relationships [59]. According to Glavas [28], these findings present a relational perspective of CSR, in which CSR inherently involves caring for stakeholders. Glavas [28] concludes from this that companies who endeavor to cultivate high-quality relationships with external stakeholders are able to create a company culture in which value is also placed on caring relationships within the organization. Wang et al.´s [31] results from the meta-study—that perceived CSR towards the public and the environment is positively correlated to OCBs—support this conclusion. Based on this assumption, we propose the hypothesis that employees in companies with high scores in indicator D2 demonstrate higher levels of OCB-I.

**Hypothesis 12 (H12).** *The scores achieved in indicator D2: cooperation with businesses in the same field are positively related to citizenship behaviors directed at co-workers.*

**4. Methods**

Figure 1 illustrates the research model of our study. In order to test the hypotheses, we work with the following model: criterion = score CGB + age + sex. In this model, differences due to corporate affiliation are contained in the CGB score. Thus, the model accommodates the nested structure of the data. Because age and sex may play a role in the

relationship between (perceived) CSR and employees' attitudes and behaviors [31], we incorporate the variables of age and sex in order to ensure the models are more exhaustive.

### 4.1. Materials

In our model, the total scores achieved in the CGBs or the percentages achieved in the indicators serve as predictors. The indicators are C1: workplace quality and affirmative action, C2: just distribution of labor, C4: just income distribution, E4: investing profits for the common good, E1: value and social impact of products and services and D2: cooperation with businesses in the same field. In order to achieve a certified CGB, a company must first of all undertake a self-assessment of how, in its own evaluation, it would perform in the indicators, (i.e., in the balance sheet). It must justify or provide evidence for its evaluations. Next, these evaluations must be validated through a peer-review process or external auditor. In the peer-review process, several companies who have undertaken a self-assessment come together and check each other's balance sheets under the professional supervision of the ECG. Otherwise, an external audit is conducted by an editor trained by the ECG. An audited CGB is valid for two years [60]. We know from interviews with 11 CgoCs in an earlier research study that, in general, only a small group of people from the companies are involved in the balance sheet process. The individuals involved are usually directors or employees with a managerial role [38].

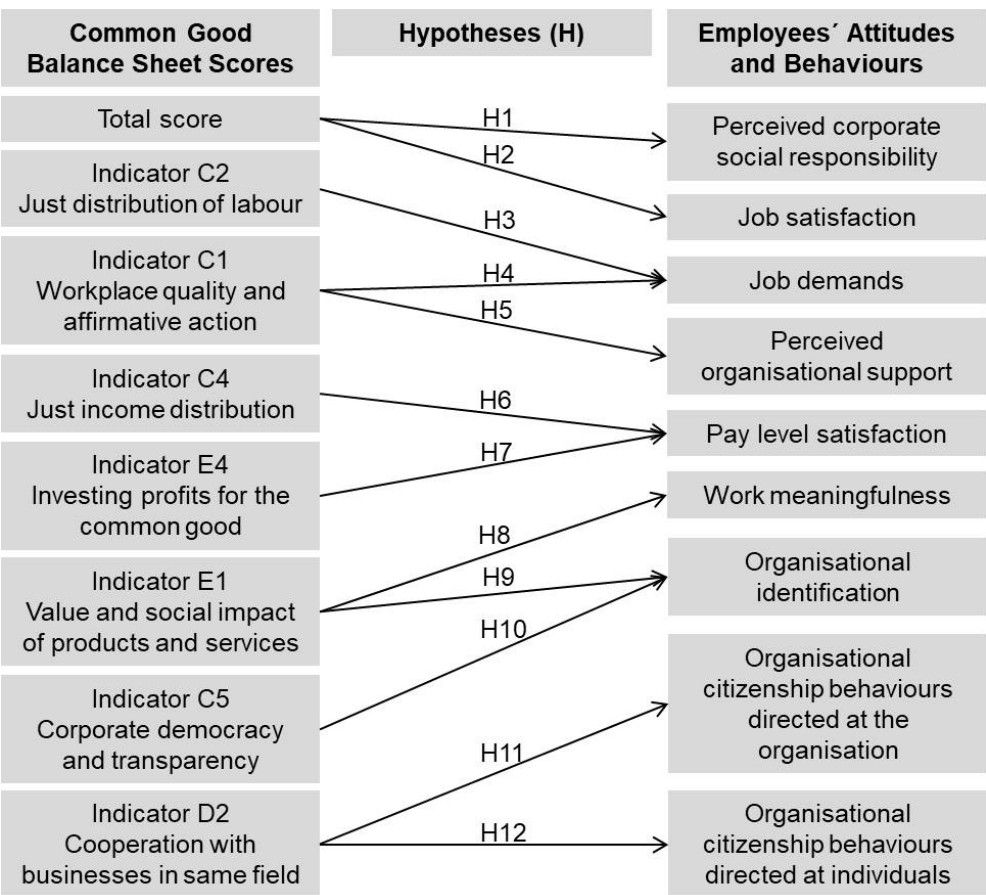

**Figure 1.** Research model of the study.

In autumn/winter 2017/18, we conducted an employee survey with eight German CgoCs who had published a CGB. Not all of these companies published an (updated) CGB after 2014. In order to ensure comparability between the companies, we need to look

at the balance sheets from the years 2012 to 2014. The balance sheets are based on the CGB that was current at the time (version 4.1).

In the survey, we collected data on the perceptions, attitudes and behavior of the employees. We used tried-and-tested scales from the literature. If no German version of the scale was available, we worked with an English native speaker to translate the scale into German using the back translation method [61]. Below, we present the scales that we used. A list of example items from the survey can be found in the Appendix A.

Perceived CSR (PCSR, $\alpha$ = 0.885): In order to measure perceived socio-ecological commitment, we chose a scale by Glavas and Kelley [25]. The scale included items pertaining to the social dimension (e.g., "contributing to the well-being of employees is a high priority in my organization") and to the ecological dimension (e.g., "environmental issues are integral to the strategy of my organization") of CSR. We recorded the responses to the items on a 5-level Likert scale from strongly disagree (1) to strongly agree (5).

Job Satisfaction ($\alpha$ = 0.861): Following recommendations by Judge and Klinger [62], we decided to measure general job satisfaction via three items. The items are (a) "All things considered, are you satisfied with your present job?"—No (1)/yes (2); (b) "How satisfied are you with your job in general?"—5-level Likert scale from very dissatisfied (1) to very satisfied (5); and (c) "Below, please write down your best estimates on the percentage of time, on average, you feel satisfied, dissatisfied, and neutral about your present job. The three figures should add up to 100%. The percentage of time I feel satisfied with my present job: _%. The percentage of time I feel dissatisfied with my present job: _%. The percentage of time I feel neutral about my present job: _%. Total:_%". For the calculation, we z-standardized the first two items as well as the information on the amount of time the individual was satisfied with their work [62]. The negative and neutral percentages were not included in the calculations. We included the neutral percentage to allow the happy and unhappy mood estimates to vary independently [63].

Job Demands (JD, $\alpha$ = 0.734): In addition to the items pertaining to general job satisfaction, we took a selection of four items pertaining to demands at work from Fischer and Lück [64]. For example, one item is, "Often, too much is expected from us at work". The responses were recorded on 5-level Likert scales from very dissatisfied (1) to very satisfied (5) or from false (1) to correct (5).

Pay Level Satisfaction (PS, $\alpha$ = 0.861): Similarly, we took the items pertaining to satisfaction with pay from Fischer and Lück [64]. Here, we asked about general pay level satisfaction as well as relative satisfaction in relation to relevant others. The response format was a 5-level Likert scale from very dissatisfied (1) to very satisfied (5).

Perceived Organizational Support (POS, $\alpha$ = 0.908): In order to determine to what extent employees feel supported by their companies, we selected the four items with the highest factor loadings from a scale published by Eisenberger et al. [65]. The items cover two aspects of POS: the extent to which the employee's contribution to the company is valued (e.g., "My company values my contribution to its well-being") and the extent to which the company cares about the employee's well-being (e.g., "My company really cares about my well-being".) The responses were recorded on a 5-level Likert scale from strongly disagree (1) to strongly agree (5).

Meaningful Work (MW, $\alpha$ = 0.916): We recorded the extent to which employees experience their work as meaningful via a multidimensional approach using nine items in total. We took these primarily from the Work as Meaning Inventory (WAMI) by Steger et al. [66]. From the subscale, "positive meaning", we selected three items (e.g., "I have found a meaningful career".) The subscale, "meaning-making through work" encompasses three items (e.g., "I view my work as contributing to my personal growth"). From the subscale, "greater good motivations" in Steger et al. [66], we selected two items. We replaced the third item from this scale, which was inverse formulated, with a positively formulated item that we found in a paper by Bunderson and Thompson [67] which states,

"what I do at work makes a difference in the world". The 5-level response Likert scale ranged from absolutely untrue (1) to absolutely true (5).

Organizational Identification (OI, $\alpha$ = 0.736): In order to record to what extent the employees identify with their company, we used a scale that was adapted to the company context and reduced by one item, a scale also used by Mael and Ashforth [53]. One example item is, "When someone criticizes my company, it feels like a personal insult". The study participants gave their responses on a 5-level Likert scale from strongly disagree (1) to strongly agree (5).

Organizational Citizenship Behaviour (OCB): In order to measure the cooperative behaviors of the employees (OCBs), we selected eight items that we found in a paper by Lee and Allen [56]. In the case of OCB-I, we decided on the items that demonstrated the highest factor loadings in the study by Zoghbi-Manrique de Lara [68]. In this study, the items asking about OCB-I ($\alpha$ = 0.775) are formulated similar to the following example: "I voluntarily take time to help others who have work-related problems". We selected the items for measuring OCB-O with the intention of recording as many aspects of the construct as possible and representing it in its full range. One example item for OCB-O ($\alpha$ = 0.789) is phrased as follows: "I take on tasks that are not required of me but that contribute to my company's image". The responses were recorded on a 5-level Likert scale from never (1) to always (5).

Demography: The study participants were asked about their biological sex, their age and their company affiliation. Sex was coded with female = 1, male = 2 and other = 3. In order to preserve the study participants' anonymity, we asked about age in seven categories.

The final questionnaire was trial-tested on 15 employees from two companies. In order to calculate the scale values from the items surveyed, each person was assigned the average value from the items on their respective scale. With respect to the calculations of the average values of the scales, we established that each person must have answered at least three items on the scale.

### 4.2. Data and Sample

The descriptive statistics for the scores in the CGBs are summarized in Table 2. As the table shows, the companies in our sample achieved total scores of around 370 to 690 points in the CGB. According to the ECG, the value of 690 points is a very good result. The ECG would expect a company that has thus far not been particularly committed to the common good to achieve between minus 100 and plus 100 points [2]. In our opinion, a CGB of around 370 points lies within the lowest spectrum of the balance sheets published thus far. Our sample therefore provides good coverage of the range of more or less CSR-committed companies with published CGBs.

**Table 2.** Descriptive statistics of scores in the common good balance sheets.

| Score | Minimum | Maximum | Mean | SD |
|---|---|---|---|---|
| CGB total (rounded to ten) | 370 | 690 | 452.84 | 81.12 |
| Indicator C1 (in %) | 20 | 79 | 42.86 | 11.41 |
| Indicator C2 (in %) | 10 | 76 | 43.09 | 13.24 |
| Indicator C4 (in %) | 20 | 80 | 47.74 | 20.80 |
| Indicator C5 (in %) | 20 | 57 | 22.19 | 7.24 |
| Indicator D2 (in %) | 30 | 73 | 41.36 | 13.71 |
| Indicator E1 (in %) | 50 | 90 | 58.98 | 8.93 |
| Indicator E4 (in %) | 10 | 100 | 91.23 | 15.74 |

CGB total = companies' total scores in the common good balance sheet (CGB); indicator C1−E4 = companies' scores in per cent in the indicators C1, C2, C5, E1 and E4 in the CGBs; SD = standard deviation.

The questionnaire was filled out online, but we also gave the companies the option of using a paper version. Two companies chose this option. In total, we acquired a data set of 378 cases. There were four cases which contained no responses to any items, so these were deleted. In addition, we deleted 11 cases in which respondents failed to provide responses after the first construct on the questionnaire. We also had to exclude 31 cases in which no company affiliation was given. The final sample composition is presented in Table 3.

**Table 3.** Study sample ($N$ = 332).

| Company's Business Field | Number of Total Employees (Rounded to Ten) | $N$ in Sample | % of Total Employees in Sample |
|---|---|---|---|
| Clothing manufacture | 470 | 116 | 35 |
| Elder care | 340 | 10 | 3 |
| Farming | 10 | 8 | 2 |
| Food production | 50 | 9 | 3 |
| Food trade I | 20 | 3 | 1 |
| Food trade II | 170 | 34 | 10 |
| Health care | 250 | 145 | 44 |
| Media production | 330 | 7 | 2 |

Of the 332 study participants, 62% were female, 36% male, one person selected "other," and 1% gave no information on their sex. The respondents ranged in age, with 30% under 35 years old, 60% between 35 and 54 years old, and 9% over 54 years old, with 1% of participants not providing any information on their age. We tried to ensure the sample was as large as possible, with employees from companies that were as diverse as possible in terms of their size and fields of activity. We already had contact with six of the companies through previous research studies [7,38], and two companies were approached by email in a second round of recruitment. In this second phase, in order to yield as large a sample as possible, we wrote only to companies with more than 100 employees. We tried to acquire further companies for the questionnaire via the ECG newsletter, however no companies responded to the call for study participants. In order to motivate companies to participate in our study and to incentivize them to encourage their employees to complete the questionnaire, we offered each company to conduct a descriptive evaluation report, in which we analyzed the answers of their employees only. The present sample is as large and diverse as we were able to make it.

*4.3. Procedure*

The statistical analysis was conducted with IBM SPSS Statistics 27. Using scatter plots, each model was tested for linear correlations both between the predictors and between the predictors and the criterion. Outliers which clearly lay more than three standard deviations below the average scale value were not taken into account in the calculations of the respective model (following the recommendation by Bühner and Ziegler [69] (p. 672 ff.). For each model, we tested the standardized residuals, Cook's distances, leverage values, Mahalanobis distances, standardized DFBetas, covariance ratios and DFFits, following the recommendations by Field [70].

The case in which the respondent selected "other" for sex was recognized as a clear outlier case in each testing. In all cases, Mahalanobis distance amounted to ≥300, where the cut-off for models with five predictors (first model variants, $p$ = 0.05) was 11.07 and the cut-off for models with 12 predictors (second model variant, $p$ = 0.05) was 21.03 (critical values of the chi-square distribution from Field [70] (p. 808). The leverage value of 0.997was clearly above the leverage threshold value of 3x [(number of predictors + 1)/(number of cases)]. This is why this case was excluded from the analyses (also from the scale reliability calculations).

We tested the residuals for normal distribution using histograms and, where necessary, also with the Kolmogorov–Smirnov and Shapiro–Wilk test. If there was no normal distribution, we followed the recommendations by Field [70] (p. 427) and used bootstrapping to generate confidence intervals and p-values. We tested whether homoscedasticity was present using scatter plots with z-standardized predicted values and standardized residuals. If necessary, we also tested this with the (modified) Breusch–Pagan test if there were high leverage values, or with the White test if there was no normal distribution in the residuals. If homoscedasticity was present, robust standard errors were predicted. In the case high leverage values were present, we used HC3 or HC4. If there was no normal distribution in the residuals, the bootstrapping results were reported (cf. recommendations by Urban and Mayerl [71], p. 279). We tested the data for multicollinearity via the variance inflation factor. Here, we followed the threshold values recommended by [70] (p. 402). We observed no problematic values in this respect.

## 5. Results

Table 4 presents the descriptive statistics and correlations of the variables used in this study. In order to test our hypotheses, multiple regressions were conducted. We controlled for employees´ age and sex in all regressions. The following section reports the results.

**Table 4.** Descriptive statistics and correlations of variables.

| V. | N | M | SD | 1 | 2 | 3 | 4 | 5 | 6 | 7 | 8 | 9 | 10 | 11 | 12 | 13 | 14 | 15 | 16 | 17 | 18 |
|---|---|---|---|---|---|---|---|---|---|---|---|---|---|---|---|---|---|---|---|---|---|
| 1 OI | 329 | 4.03 | 0.58 | | | | | | | | | | | | | | | | | | |
| 2 POS | 327 | 3.56 | 0.92 | **0.524** | | | | | | | | | | | | | | | | | |
| 3 WM | 331 | 3.45 | 0.76 | **0.606** | **0.581** | | | | | | | | | | | | | | | | |
| 4 JS | 320 | 0.01 | 0.89 | **0.446** | **0.657** | **0.591** | | | | | | | | | | | | | | | |
| 5 JD | 330 | 2.60 | 0.83 | **−0.229** | **−0.554** | **−0.351** | **−0.563** | | | | | | | | | | | | | | |
| 6 PS | 325 | 3.05 | 1.12 | **0.258** | **0.355** | **0.312** | **0.330** | **−0.246** | | | | | | | | | | | | | |
| 7 OCB-O | 327 | 3.64 | 0.72 | **0.456** | **0.300** | **0.483** | **0.266** | -0.137 * | 0.123 * | | | | | | | | | | | | |
| 8 OCB-I | 330 | 4.23 | 0.57 | **0.315** | **0.202** | **0.239** | **0.231** | -0.119 * | **0.145** | **0.376** | | | | | | | | | | | |
| 9 PCSR | 328 | 4.22 | 0.57 | **0.373** | **0.573** | **0.454** | **0.368** | **−0.323** | 0.115 * | **0.307** | **0.196** | | | | | | | | | | |
| 10 Age | 328 | 3.29 | 1.60 | 0.141 * | −0.005 | 0.103 | 0.026 | 0.053 | **0.180** | **0.150** | −0.106 | −0.067 | | | | | | | | | |
| 11 Sex | 327 | 1.37 | 0.48 | 0.081 | 0.053 | 0.048 | 0.127 * | -0.051 | 0.092 | **0.185** | 0.016 | −0.007 | 0.138 * | | | | | | | | |
| 12 CGB | 331 | 453 | 81 | 0.001 | **0.239** | **0.183** | 0.104 | **−0.170** | −0.100 | 0.169 | −0.135 * | **0.317** | 0.100 | 0.034 | | | | | | | |
| 13 C1 | 331 | 43 | 11 | 0.097 | **0.288** | 0.102 | 0.136 | −0.238 | −0.007 | 0.039 | 0.046 | 0.180 | −0.047 | −0.034 | **0.457** | | | | | | |
| 14 C2 | 331 | 43 | 13 | 0.132 * | 0.136 * | 0.031 | 0.081 | -0.143 * | **0.183** | −0.041 | **0.194** | −0.101 | −0.041 | −0.015 | **−0.165** | **0.687** | | | | | |
| 15 C4 | 331 | 48 | 21 | 0.099 | −0.086 | 0.087 | 0.013 | 0.058 | **0.378** | 0.025 | 0.086 | **−0.255** | **0.179** | 0.073 | **−0.315** | **−0.431** | 0.107 | | | | |
| 16 E4 | 331 | 91 | 16 | 0.086 | −0.043 | −0.026 | 0.024 | 0.025 | 0.072 | 0.019 | 0.223 | 0.016 | −0.102 | −0.022 | **−0.242** | **0.171** | **0.321** | 0.035 | | | |
| 17 E1 | 331 | 59 | 9 | 0.082 | 0.028 | **0.155** | 0.058 | −0.002 | **0.320** | 0.119 * | 0.043 | −0.042 | **0.234** | 0.134 * | **0.227** | −0.243 | −0.014 | **0.777** | −0.116 * | | |
| 18 D2 | 331 | 41 | 14 | 0.077 | 0.034 | **0.188** | 0.065 | 0.001 | **0.276** | **0.148** | −0.020 | 0.015 | **0.248** | 0.113 * | **0.330** | **−0.306** | **−0.283** | **0.715** | −0.119 * | **0.904** | |
| 19 C5 | 331 | 22 | 7 | 0.113 * | **0.216** | **0.230** | 0.122 * | **−0.193** | **0.200** | 0.116 * | 0.010 | 0.015 | **0.175** | 0.036 | **0.511** | **0.515** | **0.454** | **0.320** | **−0.306** | **0.496** | **0.445** |

Bold correlations are significant at the 0.01 level (2-tailed). * Correlation is significant at the 0.05 level (2-tailed). V. = Variable, OI = organizational identification, POS = perceived organizational support, WM = work meaningfulness, JS = job satisfaction (z-standardized values), JD = job demands, PS = pay level satisfaction, OCB-O = organizational citizenship behaviors directed at the organization, OCB-I = organizational citizenship behaviors directed at individuals, PCSR = perceived corporate social responsibility, CGB = total score in the common good balance, C1 = indicator score C1, C2 = indicator score C2, C4 = indicator score C4, E4 = indicator score E4, E1 = indicator score E1, D2 = indicator score D2, C5 = indicator score C5.

### 5.1. Perceived CSR and Job Satisfaction

The regression to predict employee´s perception of CSR based on the company´s total score in the CGB is significant ($F_{(4, 318)}$ = 11.912, $p < 0.000$, $1 - ß = 0.999$), with an $R^2$ of 0.130. Regression coefficients are shown in Table 5. The results indicate a positive association between the total scores in the CGB and employees' perception of CSR (b = 0.002 [0.002, 0.003]; $p = 0.000$), supporting Hypothesis 1.

**Table 5.** Results from multiple linear regression testing Hypothesis 1.

| Criterium | | PCSR (H1) Robust SD (HC3), *N* = 323 | | | | |
|---|---|---|---|---|---|---|
| **Equation** | **R²** | **Adj. R²** | **b** | **SE B** | **ß** | ***p*** |
| Score CGB + Age + Sex | 0.130 | 0.119 | | | | 0.000 |
| **Predictor** | | | | | | |
| Constant | | | 3.266 (2.930, 3.601) | 0.171 | | 0.000 |
| Score CGB | | | 0.002 (0.002, 0.003) | 0.000 | 0.345 | 0.000 |
| 35–54 years | | | −0.132 (−0.253, −0.012) | 0.061 | −0.118 | 0.032 |
| >54 years | | | −0.114 (−0.352, 0.123) | 0.121 | −0.061 | 0.344 |
| Men | | | 0.025 (−0.094, 0.144) | 0.060 | 0.022 | 0.678 |

95% confidence intervals reported in parentheses, CGB = common good balance sheet, PCSR = perceived corporate social responsibility.

The regression to predict employee's job satisfaction based on the companies' total scores in the CGB showed a barely significant result ($F_{(4, 312)}$ = 2.441, $p < 0.047$, $1 - ß = 0.699$), with an $R^2$ of 0.030. Regression coefficients are shown in Table 6. The results indicate a positive association between the total scores in the CGB and overall job satisfaction (b = 0.001 [0.000, 0.002]) with $p < 0.058$. When testing one-sided the result is statistically significant. Accordingly, employees who work for a company with higher total scores in the CGB report higher levels of job satisfaction, thus supporting Hypothesis 2.

**Table 6.** Results from multiple linear regressions testing Hypothesis 2.

| Criterium | | JS (H2) Confidence Intervals, Standard Errors and PS Based on 1000 Bootstrap Samples. *N* = 317 | | | | |
|---|---|---|---|---|---|---|
| **Equation** | **R²** | **Adj. R²** | **b** | **SE B** | **ß** | ***p*** |
| Score CGB + Age + Sex | 0.030 | 0.018 | | | | 0.047 |
| **Predictor** | | | | | | |
| Constant | | | −0.549 (−1.066, 0.008) | 0.284 | | 0.054 |
| Score CGB | | | 0.001 (0.000, 0.002) | 0.001 | 0.101 | 0.058 |
| 35–54 years | | | −0.041 (−0.253, 0.179) | 0.109 | −0.023 | 0.708 |
| >54 years | | | 0.108 (−0.266, 0.438) | 0.186 | 0.034 | 0.564 |
| Men | | | 0.183 (0.026, 0.431) | 0.099 | 0.128 | 0.016 |

95% bias corrected and accelerated confidence intervals reported in parentheses, CGB = common good balance sheet, JS = job satisfaction.

### 5.2. Job Demands and Perceived Organisational Support

Regressions are significant to predict employees' perceived job demands based on the companies' indicator score, C2: just distribution of labor in the CGB ($R^2 = 0.030$, $F_{(4321)}$ = 2.461, $p < 0.045$, $1 - ß = 0.731$) and based on the indicator scores C1: workplace quality and affirmative action ($R^2 = 0.068$, $F_{(4321)}$ = 5.860, $p < 0.000$, $1 - ß = 0.984$). Regression coefficients are shown in Tables 7 and 8. The results indicate a negative association between indicator score C2 (b = −0.009 [−0.016, −0.002], $p = 0.009$) as well as indicator score C1 (b = −0.018 [−0.026, −0.010], $p = 0.000$) and employees' perceived demands. Hence, employees who work for a company with higher scores in the indicator C2 and C1 in the CGB report less job demands. These results support Hypothesis 3 and 4.

**Table 7.** Results from multiple linear regression testing Hypothesis 3.

| Criterium | | | JD (H3) $N = 326$ | | | |
|---|---|---|---|---|---|---|
| Equation | R² | Adj. R² | b | SE B | ß | p |
| Score C2 + Age + Sex | 0.030 | 0.018 | | | | 0.045 |
| **Predictor** | | | | | | |
| Constant | | | 2.933 (2.587, 3.279) | 0.176 | | 0.000 |
| Score C2 | | | −0.009 (−0.016, −0.002) | 0.004 | −0.146 | 0.009 |
| 35–54 years | | | 0.159 (−0.044, 0.361) | 0.103 | 0.093 | 0.124 |
| >54 years | | | 0.024 (−0.321, 0.369) | 0.175 | 0.008 | 0.891 |
| Men | | | −0.114 (−0.302, 0.074) | 0.096 | −0.066 | 0.234 |

95% confidence intervals reported in parentheses, JD = job demands.

**Table 8.** Results from multiple linear regression testing Hypothesis 4.

| Criterium | | | JD (H4) $N = 326$ | | | |
|---|---|---|---|---|---|---|
| Equation | R² | Adj. R² | b | SE B | ß | p |
| Score C1 + Age + Sex | 0.068 | 0.056 | | | | 0.000 |
| **Predictor** | | | | | | |
| Constant | | | 3.310 (2.932, 3.689) | 0.193 | | 0.000 |
| Score C1 | | | −0.018 (-0.026, −0.010) | 0.004 | −0.244 | 0.000 |
| 35–54 years | | | 0.151 (−0.047, 0.349) | 0.101 | 0.089 | 0.134 |
| >54 years | | | 0.015 (−0.323, 0.352) | 0.171 | 0.005 | 0.932 |
| Men | | | −0.124 (−0.308, 0.061) | 0.094 | −0.072 | 0.187 |

95% confidence intervals reported in parentheses, JD = job demands.

The regression to predict employees' perceived organizational support based on the indicator scores, C1: work place quality and affirmative action is also significant ($F_{(4, 317)} = 9.642$, $p < 0.000$, $1 − ß = 0.999$), with an R² of 0.108. Regression coefficients are shown in Table 9. The results indicate a positive association between indicator score C1 and employees' perception of organizational support (b = 0.025 [0.016, 0.034]; $p = 0.001$). Therefore, employees who work for a company with higher scores in indicator C1 in the CGB report higher levels of perceived organizational support, which supports Hypothesis 5.

**Table 9.** Results from multiple linear regression testing Hypothesis 5.

| Criterium | | | POS (H5) Confidence Intervals, Standard Errors and PS Based on 1000 Bootstrap Samples. $N = 322$ | | | |
|---|---|---|---|---|---|---|
| Equation | R² | Adj. R² | b | SE B | ß | p |
| Score C1 + Age + Sex | 0.108 | 0.097 | | | | 0.000 |
| **Predictor** | | | | | | |
| Constant | | | 2.481 (2.066, 2.869) | 0.219 | | 0.001 |
| Score C1 | | | 0.025 (0.016, 0.034) | 0.004 | 0.313 | 0.001 |
| 35–54 years | | | −0.113 (−0.318, 0.111) | 0.099 | −0.061 | 0.253 |
| >54 years | | | 0.203 (−0.180, 0.563) | 0.187 | 0.062 | 0.288 |
| Men | | | 0.160 (−0.055, 0.372) | 0.098 | 0.085 | 0.105 |

95% bias corrected and accelerated confidence intervals reported in parentheses, POS = perceived organizational support.

*5.3. Pay Level Satisfaction*

The regression to predict employees' pay level satisfaction based on the companies' indicator scores, C4: just income distribution in the CGB is significant ($F_{(4, 316)} = 14.530$, $p < 0.000$, $1 − ß = 0.999$), with an R² of 0.155. Regression coefficients are shown in Table 10.

The results show a positive association between the indicator scores C4 and employees′ pay level satisfaction (b = 0.019 [0.013, 0.025]; $p$ = 0.001). Accordingly, employees who work for a company with higher score in the indicator C4 in the CGB report higher levels of pay level satisfaction, thus supporting Hypothesis 6.

**Table 10.** Results from multiple linear regression testing Hypothesis 6.

| Criterium | PS (H6) Confidence Intervals, Standard Errors and PS Based on 1000 Bootstrap Samples. *N* = 321 | | | | | |
|---|---|---|---|---|---|---|
| **Equation** | **R²** | **Adj. R²** | **b** | **SE B** | **ß** | **p** |
| Score C4 + Age + Sex | 0.155 | 0.145 | | | | 0.000 |
| **Predictor** | | | | | | |
| Constant | | | 1.906 (1.615, 2.200) | 0.152 | | 0.001 |
| Score C4 | | | 0.019 (0.013, 0.025) | 0.003 | 0.354 | 0.001 |
| 35–54 years | | | 0.281 (0.034, 0.525) | 0.121 | 0.123 | 0.024 |
| >54 years | | | 0.281 (−0.264, 0.820) | 0.256 | 0.072 | 0.275 |
| Men | | | 0.107 (−0.130, 0.350) | 0.124 | 0.046 | 0.391 |

95% bias corrected and accelerated confidence intervals reported in parentheses, PS = pay level satisfaction.

Further, we find a significant regression to predict employees' pay level satisfaction based on the companies' indicator scores, E4:investing profits for the common good in the CGB ($n$ = 321, $F_{(4, 316)}$ = 3.807, $p$ < 0.005, 1 − ß = 0.896), with an $R^2$ of 0.046 (adjusted $R^2$ = 0.034). The results indicate no association between the indicator scores E4 and employees′ pay level satisfaction (b = 0.007 [−0.001, 0.018], SE B = 0.005, ß = 0.096, $p$ = 0.125) nor between indicator score E4 and sex (men: b = 0.153 [−0.100, 0.391], SE B = 0.132, ß = 0.066, $p$ = 0.239). Although the results show positive relations between pay level satisfaction and age (35–54 years: b = 0.361 [0.076, 0.619], SE B = 0.138, ß = 0.158, $p$ = 0.008 and >54 years: b = 0.655 [0.166, 1.131], SE B = 0.244, ß = 0.168, $p$ = 0.010), our data do not support Hypothesis 7.

*5.4. Meaningful Work*

The regression to predict employees′ perception of work meaningfulness based on the companies' indicator score, E1: value and social impact of products and services in the CGB is significant ($F_{(4, 321)}$ = 2.879, $p$ < 0.023, 1 − ß = 0.789), with an $R^2$ of 0.035. Regression coefficients are shown in Table 11. The results indicate a positive association between the scores in the indicator E1 in the CGB and employees' perception of work meaningfulness (b = 0.011 [0.001, 0.020]; $p$ = 0.022). Accordingly, employees who work for a company with higher indicator scores E1 in the CGB report higher levels of work meaningfulness, thus supporting Hypothesis 8.

**Table 11.** Results from multiple linear regression testing Hypothesis 8.

| Criterium | WM (H8) Confidence Intervals, Standard Errors and PS Based on 1000 Bootstrap Samples. *N* = 326 | | | | | |
|---|---|---|---|---|---|---|
| **Equation** | **R²** | **Adj. R²** | **b** | **SE B** | **ß** | **p** |
| Score E1 + Age + Sex | 0.035 | 0.023 | | | | 0.023 |
| **Predictor** | | | | | | |
| Constant | | | 2.746 (2.218, 3.289) | 0.284 | | 0.001 |
| Score E1 | | | 0.011 (0.001, 0.020) | 0.005 | 0.129 | 0.022 |
| 35–54 years | | | 0.062 (−0.106, 0.245) | 0.088 | 0.040 | 0.479 |
| >54 years | | | 0.269 (−0.069, 0.580) | 0.162 | 0.103 | 0.095 |
| Men | | | 0.025 (−0.153, 0.197) | 0.088 | 0.016 | 0.783 |

95% bias corrected and accelerated confidence intervals reported in parentheses, WM = work meaningfulness.

### 5.5. Organisational Identification

We also find a significant regression to predict employees' organizational identification based on the indicator score E1 ($n$ = 323, F (4, 318) = 2.802, $p$ < 0.026, 1 − ß = 0.770), with an $R^2$ of 0.034 (adjusted $R^2$ = 0.022, $p$ = 0.026). However, the results indicate no association between the indicator score E1 and employees' organizational identification (b = 0.004 [−0.003, 0.012], SE B = 0.004, ß = 0.068, $p$ = 0.242) nor between indicator score E1 and sex (men: b = 0.069 [−0.060, 0.197], SE B = 0.065, ß = 0.059, $p$ = 0.293) nor between indicator score E1 and age in respect to older employees (>54 years: b = 0.206 [−0.039, 0.452], SE B = 0.125, ß = 0.104, $p$ = 0.099). Although the results show a positive relation between organizational identification and middle aged employees (35–54 years: b = 0.159 [0.023, 0.295], SE B = 0.069, ß = 0.139, $p$ = 0.023), our data do not support Hypothesis 9.

However, the regression to predict employees′ organizational identification based on the companies′ indicator scores, C5: corporate democracy and transparency in the CGB is significant (F (4, 318) = 3.296, $p$ < 0.011, 1 − ß = 0.845), with an $R^2$ of 0.040. Regression coefficients are shown in Table 12. The results indicate a positive association between indicator scores C5 and employees' organizational identification (b = 0.008 [−0.002, 0.016]) with $p$ = 0.052. When testing one-sided the result is statistically significant. Accordingly, employees who work for a company with higher scores in the indicator C5 in the CGB report higher levels of organizational identification, thus supporting Hypothesis 10.

**Table 12.** Results from multiple linear regression testing Hypothesis 10.

| Criterium | | | OI (H10) Confidence Intervals, Standard Errors and PS Based on 1000 Bootstrap Samples. *N* = 323 | | | |
|---|---|---|---|---|---|---|
| Equation | $R^2$ | Adj. $R^2$ | b | SE B | ß | p |
| Score C5 + Age + Sex | 0.040 | 0.028 | | | | 0.011 |
| **Predictor** | | | | | | |
| Constant | | | 3.729 (3.533, 3.929) | 0.104 | | 0.001 |
| Score C5 | | | 0.008 (−0.002, 0.016) | 0.004 | 0.102 | 0.052 |
| 35–54 years | | | 0.154 (0.018, 0.291) | 0.071 | 0.134 | 0.040 |
| >54 years | | | 0.201 (−0.052, 0.460) | 0.129 | 0.101 | 0.120 |
| Men | | | 0.076 (−0.045, 0.197) | 0.060 | 0.065 | 0.215 |

95% bias corrected and accelerated confidence intervals reported in parentheses, OI = organizational identification.

### 5.6. Organisational Citizenship Behaviours

The regression to predict employees' organizational citizenship behaviors towards the company (OCB-O) based on the companies' indicator scores, D2—cooperation with businesses in the same field is significant (F (4, 317) = 6.825, $p$ < 0.000, 1 − ß = 0.994), with an $R^2$ of 0.079. Regression coefficients are shown in Table 13. The results indicate a positive association between indicator score D2 and OCB-O (b = 0.005 [−0.001, 0.011]; $p$ = 0.088). When testing one-sided, the result is statistically significant. Thus, employees who work for a company with higher scores in indicator D2 in the CGB report higher levels of OCB-O, which supports Hypothesis 11.

**Table 13.** Results from multiple linear regression testing Hypothesis 11.

| Criterium | | | OCB-O (H11) *N* = 322 | | | |
|---|---|---|---|---|---|---|
| Equation | R² | Adj. R² | b | SE B | ß | p |
| Score D2 + Age + Sex | 0.079 | 0.068 | | | 0.000 | |
| **Predictor** | | | | | | |
| Constant | | | 3.273 (3.005, 3.542) | 0.136 | | 0.000 |
| Score D2 | | | 0.005 (−0.001, 0.011) | 0.003 | 0.100 | 0.088 |
| 35–54 years | | | 0.056 (−0.112, 0.225) | 0.086 | 0.039 | 0.510 |
| >54 years | | | 0.419 (0.115, 0.723) | 0.155 | 0.170 | 0.007 |
| Men | | | 0.231 (0.074, 0.389) | 0.080 | 0.158 | 0.004 |

95% confidence intervals reported in parentheses, OCB-O = organizational citizenship behaviors directed at the organization.

However, the regression to predict employees' organizational citizenship behaviors towards individuals based on the indicator scores D2 is not significant (*n* = 324, F (4, 319) = 1.656, *p* < 0.160, 1 − ß = 0.506), with an R² of 0.020 (adjusted R² = 0.008, *p* = 0.160). Thus, our data do not support Hypothesis 12.

**6. Discussion**

The scores from the CGBs give us an objective measurement as a predictor of employees' attitudes and behaviors. This is relatively unusual in micro-CSR research. As Jones et al. [72] have established, CSR in predominantly survey-based micro-CSR research is almost always operationalized through measurements of employees' perceptions or beliefs about their employer's CSR practices. One of the reasons given for this is that employees do not typically respond to CSR practices as they objectively exist, but to CSR practices as they perceive them to exist [28,30,54]. The extent to which employees perceive CSR is also part of our study. The primary aim of our study, however, was to explore whether there were any correlations between a better performance in the CGB and job-related attitudes and employee behavior. The first part of the discussion of our results is divided into the constructs under study. We also indicate here the possibilities for further research. In the second part of the discussion, we address the limitations of our study.

*6.1. Perceived CSR*

According to our results, the values achieved in the CGBs correlate positively with perceived CSR. Employees from companies with higher values in the CGBs indicate that they perceive more CSR. This result suggests that companies with higher values in the CGBs are more committed to the common good, not only in the eyes of the ECG evaluators but also in the eyes of their employees.

*6.2. Job Satisfaction*

Some CgoCs from the study by Sanchis et al. [22] have reported that participation and communication within their companies have improved since their first CGB, and that employees' commitment, motivation and satisfaction have increased. According to Ollé-Espluga et al. [20], CgoCs provide more favorable conditions in terms of training and participation, as well as in terms of control and flexibility regarding working hours and place of work, in comparison with the Austrian and German economy overall. Since these aspects play a role in the evaluation of CGBs, it should also be the case that companies with higher values in the CGBs also offer better working conditions, leading to greater job satisfaction.

Our results show that the values achieved in the CGBs correlate positively with job satisfaction. The employees from companies with higher scores in the CGBs are more

satisfied with their jobs than employees from companies that achieve fewer points in the CGB. Hence, the conditions that lead to more job satisfaction increase in companies with higher points in the CGBs. However, we only studied companies that have published a CGB. That is why we do not know how the job satisfaction in CgoCs differs from that in companies without a CGB. Hence, to provide some context, a follow-up comparative study on job satisfaction in companies both with and without CGBs would be useful.

Nonetheless, the score in the CGB correlates positively with job satisfaction. This means that, assuming better working conditions lead to greater job satisfaction, we may conclude that the CGB is a suitable tool for a comparative evaluation of working conditions in CgoCs. However, it must be noted that the total score in the CGB is added together from the results of the individual indicators, so it is theoretically also possible to achieve a positive result in the CGB purely through a high external CSR or purely through CSR towards the environment. Hence, the total score in an individual case does not necessarily say anything about the working conditions in the company, and when in doubt, it should be checked against the scoring in the individual indicators.

In the overall sample, job satisfaction is evaluated as medium, with a slight positive tendency (M = 0.008, z-standardized value), and here we observe considerable variance (SD = 0.887). This means that there are employees in the companies who are very satisfied with their job but also employees who are very dissatisfied with their work. Considering only a small percentage of the variance in job satisfaction is explained by the score in the CGB ($R^2$ = 0.030), the question remains open as to which factors significantly determine job satisfaction in the CgoCs. Even if the CGB provides for a certain level of comparability with regard to working conditions, it does not seem to include the relevant factors that determine job satisfaction. In order to identify these missing factors, we could consult theories on job satisfaction, such as Hackman and Oldham [73] job diagnostic survey, and absorb into the indicators, where appropriate, factors influencing job satisfaction that have not yet been considered. Furthermore, it is not only the objective measures implemented by a company for internal and external stakeholders that are relevant to job satisfaction. Subjective evaluations by the employees should be incorporated into the CGO evaluation of a company and therefore into the CGB score. Thus far, merely conducting company employee surveys has had a positive impact on the balance sheet, while the actual result of the surveys has not had any bearing.

*6.3. Job Demands and Perceived Organisational Support*

The ECG aims to establish a "just" distribution of working hours: its objective is to reduce regular weekly working hours [16]. Our study shows that the percentage achieved in indicator C2: just distribution of labor is negatively correlated with the demands in the workplace. This means that as CGO increases with regard to the distribution of labor, the demands in the workplace decrease. The fact that as CGO increases, the demands in the workplace decrease, could explain why job satisfaction increases in CgoCs with increasing CGO. It is worth noting, however, that according to Karasek's [74] job demands-job control model, there is no simple linear relationship between job demands and job satisfaction; instead, there is an interaction effect in which control in the workplace has a role to play. It is only where there are high demands coupled with a low decision latitude that the result is dissatisfaction in the workplace [74]. Since the working conditions in CgoCs are characterized by participation, control and flexibility [20], job satisfaction may even be high among employees who are subject to high demands. Hence, job demands may play a secondary role with regard to job satisfaction in CgoCs. The correlations would therefore need to be explored in further empirical studies.

The criteria of workplace quality and equal opportunity may be of relevance in explaining the increasing job satisfaction in companies with higher CGO. In the explorative interview study by Meynhardt and Fröhlich [21], CgoCs reported that compiling the CGB not only created an awareness of the need to reduce job demands but also raised awareness around interacting with employees, women in leadership roles, disabled access

within the workplace, diversity of opinion, as well as the introduction of a behavioral code. These criteria are represented by the indicator C1: workplace quality and affirmative action, which correlates negatively with the demands in the workplace. Hence, if a company increasingly champions workplace quality and equal opportunity, the job demands decrease. We propose the hypothesis that job demands decrease with increasing workplace quality, and that an interplay between both factors leads to increased job satisfaction. It is similar with POS. The indicator C1 correlates positively with the support the employees perceive the company offers them. Where there is increasing workplace quality and equal opportunity, the employees also feel increasingly supported. We assume there is a correlation between the increasing scores in indicator C1, the resulting increase in POS, and job satisfaction, which increases as the total scores in the CGB increase. Follow-up studies are needed in order to explain the nature of these correlations more precisely.

### 6.4. Pay Level Satisfaction

According to Lawler's model of job satisfaction, an employee should be satisfied with their income if the amount of perceived income corresponds to the income they feel they are entitled to [48]. Here, processes of social comparison, e.g., with friends and colleagues, have a role to play [48]. The ECG requires that companies demonstrate "just" and transparent distribution of income and profits. According to our results, the percentages achieved in indicator C4: just income distribution correlate positively with satisfaction with pay. We may therefore assume that the employees in the CgoCs share the ECG's concepts of justice with regard to income distribution, and that, according to Lawler's model, increased application of these concepts and a simultaneous increase in transparency in the company lead to rising pay level satisfaction.

However, there is no apparent correlation between the scores achieved in indicator E4: investing profits for the common good and satisfaction with pay. This would mean that profit distribution within the company is of no relevance to pay level satisfaction. However, it is striking that the average within the indicator is very high (M = 91%) and, at the same time, the standard deviation is low (SD = 16%). Hence, it is predominantly companies with a very high value in indicator E4 that are part of our sample. This variance restriction may lead to an underestimation of the correlations [69], which means our result is of a provisional nature. In order to determine the influence of profit distribution on pay level satisfaction, it would be advisable to conduct a comparative study with CgoCs and companies that would achieve only very few points in this indicator.

Further studies on pay and pay satisfaction would be interesting too. Some research questions could be, how high is the pay in CgoCs in comparison with "conventional" companies? In which companies are the employees more satisfied with their pay? The absolute pay level has only a slight influence on satisfaction with pay [47]. As the results of our study show, a fair and transparent distribution of wages may go some way to explaining pay level satisfaction. We assume that pay is relatively low in CgoCs because of extra expenditure on their socio-ecological economic activities. It would be interesting to establish to what extent CgoCs are able to "compensate" for their potentially relatively low pay through fair distribution and transparency.

### 6.5. Meaningful Work

The results of our study show that the percentages achieved in indicator E1: value and social impact of products and services correlate positively with the employees' experience of the meaningfulness of their work. As the social relevance of the products and services of companies increases, so does the employees' perception of their work as meaningful. In our interpretation, the result shows an intersection between what the ECG regards as a company's meaningful contribution to society and what the employees consider meaningful, i.e., what their company contributes to the world. This confirms the assumption by Rosso et al. [36] that companies' emphasis on their contribution to the

common good may have positive implications for employees' experience of meaningfulness. The experience of meaningful work correlates strongly with job satisfaction [50], which means that increasing meaningfulness of work as a result of the increasing meaningfulness of products and services may also be connected with increasing job satisfaction where there is a higher CGO within the company. This hypothetical correlation would, however, need to be tested. Neither is it clear whether, in general, employees from CgoCs experience more meaningfulness at work than employees from companies without a CGB. This, too, would be a question for a possible follow-up study.

Glavas and Kelley [25] have established that the mechanisms through which employee perceptions of CSR impact their work behaviors and attitudes are still, to a large extent, unclear. Our study is unable to provide empirical information about the processes underlying the correlations that have been established. The model on the correlation between the meaningfulness and social impact of products and services and the perceived meaningfulness of work has only low explanatory power ($R^2 = 0.035$), so only a small percentage of the variance in meaningful work is explained by the meaningfulness of products and services. According to Glavas and Kelley [25], employees' sense of meaningfulness is increased by perceived CSR only when actions are directed towards third parties and not in terms of how the organization treats the employees themselves. A better elucidation of the variance in meaningful work might be possible if we also include in the model the other indicators that record external CSR (e.g., indicator E2: contribution to the local community). This could be tested in future studies and could immediately provide information on which aspects of CGO are of particular relevance in the experience of meaningful work. We already know that values and value congruence have a role to play in this connection and are also important in the processes whereby CSR impacts on employees [28,50]. Due to its explicit reference to values, the ECG lends itself to deeper research on the significance of values and value congruence in the field of micro-CSR.

*6.6. Organisational Identification*

Unlike in the case of the perceived meaningfulness of work, we are unable to find a correlation between the percentages achieved in indicator E1: value and social impact of products and services and organizational identification. In the meta-study by Wang et al. [31], CSR, internal CSR and external CSR correlated positively with organizational identification. If we stay with the definition of CSR as a company caring for the well-being of its employees and other key stakeholders, including the societal and natural environment with the aim of also creating value for the business, the creation of meaningful and socially-relevant products and services can also be understood as external CSR. This means our findings are not consistent with the research results thus far on the correlation between CSR and organizational identification. According to John et al. [55], employees who evaluate the CSR of their company positively should be proud of their company, and this in turn should lead through a self-categorization process, to greater organizational identification. Here, CSR conveys to the employees a value fit between the companies and the employees, and the employees derive organizational identification from this [31]. It is therefore surprising that we were unable to find any correlation between the meaning and social impact of the products and services and organizational identification among the employees from the CgoCs, especially as the companies report that by publishing CGBs they find employees who are a better fit and who share their values [24]. It is possible, however, that we found no correlation here either because, as with common good-oriented income distribution, there is a "ceiling effect". All the companies in the sample supply products that are geared towards basic needs, i.e., food production and trade, health and elderly care, clothing and political media, and that are therefore more likely to be evaluated as meaningful and socially relevant. This is also reflected in the descriptive statistics for indicator E1 (minimum = 50%, maximum = 90%, M = 58.98% and SD = 8.93%). That is why a study that includes companies that perform considerably

worse in indicator E1 than the companies in the present sample has the potential to show up effects.

Although Wang et al. [31] report that internal CSR correlates positively with organizational identification, Wang and colleagues indicate, at the same time, that the relationship between organizational identification and perceived CSR towards employees was not significant. According to our results, the percentage achieved in indicator C5: corporate democracy and transparency correlates positively with the employees' identification with their companies. We assume that employees' feelings of belonging to their companies are strengthened by the extensive transparency and employee participation in fundamental decision-making processes required by the ECG, and that employees can therefore identify more with their companies [31]. At the same time, including employees in important decision-making processes increases self-esteem, which in turn leads to more organizational identification. According to organizational identity theory [51], employees tend to identify with organizations from which they can derive self-esteem and self-respect [31]. Internal CSR, or rather CSR towards employees, connected with transparency and participation therefore has a positive influence on organizational identification.

### 6.7. Organisational Citizenship Behaviours

Some CgoCs report improvements in cooperation strategies among businesses and better relations with suppliers since conducting their first CGB [22]. We observed that as the percentage increases in indicator D2: cooperation with businesses in the same field, marking the company's stronger cooperation with other companies, the employees report more OCBs on their part towards their company. According to Glavas [28], this finding can be explained by the fact that if a company goes above and beyond its primary tasks and aims to contribute to the greater good of society, then employees will go above and beyond their primary tasks to contribute to the greater good of the organization. This explanation can only be correct if the employees perceive that their companies really are behaving cooperatively. The result of our study may therefore suggest that the CGBs do in fact represent the variability in the cooperative behavior of the companies, that the extent of cooperative behavior is also perceived by the employees and that more cooperative behavior ultimately results in more OCB.

By contrast, there is no confirmation of our assumption that increasing cooperative behavior between companies results in employees demonstrating increasing OCBs towards their colleagues. We did not observe cooperative behavior by companies having a spillover effect upon the individual behavior of particular employees. This is not consistent with the results from the meta-study by Wang et al. [31], according to which perceived CSR towards the public and the environment is positively correlated to OCBs, which led us to assume that CSR or cooperative behavior towards other companies would likewise lead to more OCBs. Neither, therefore, do our results give support to Glavas' [28] hypothesis that companies that endeavor to create high-quality relationships with external stakeholders thereby create a company culture in which caring relationships are important within the organization.

### 6.8. Limitations

Unfortunately, we were unable to discuss the differences with respect to age and sex in greater depth within the scope of this paper. Further studies could explore the effects of age and sex in more detail. Considering there was one case where the sex was given as "other," this was a clear outlier in all models so we excluded this case from our analyses. However, we were unable to decide whether the reason we saw such strong deviation in the responses was because the questionnaire was not filled out "seriously" or whether the person in question really does demonstrate considerably divergent feelings and behavior that might also be connected with their gender identity. Follow-up studies could therefore focus on gender identity and its role in the context of micro-CSR.

Neither were we able to rule out gender bias, for two thirds of the respondents were female. According to the study by Ollé-Espluga et al. [20], women represent almost half of the workforce in ECG firms. However, according to the valid frequency, women account for two thirds of the total workforce in their study [20]. Hence, it could certainly also be the case that significantly more women than men work in CgoCs and that our study therefore represents the gender ratio correctly.

Unluckily, not all companies compiled a CGB in the same year. In addition, the majority of companies have not published an updated CGB since 2014. This means that in order to ensure the highest possible level of comparability, we used the balance sheets from the years 2012 to 2014. Even if a balance sheet is valid for two years, we cannot rule out limitations in comparability. In addition, the validity of our study may also be compromised by the fact that several years separate the balance sheets and the employee survey.

Furthermore, we observed that Mahalanobis distances were often higher than the threshold values. For example, we observed with the calculations for Hypothesis 10 that for the first eight cases in the data set, all the values were around the value 28, whereas the threshold value in the case of five predictors amounted to 11.07 ($p = 0.05$). All these cases were from the same company. Due to the frequent appearance of cases with these relatively small deviations, we did not exclude these from the analyses. We only deleted very noticeable outliers, e.g., with values higher than 300, from the respective analyses. We are therefore unable to rule out possible distortions in our calculations.

Last but not least, we would like to point out that there were nine occasions on which the respondent stopped filling in the questionnaire when they arrived at the item, "My work helps me understand myself better". Since these dropouts were noticeably frequent, we cannot recommend using this item.

## 7. Conclusions

The CGB is a CSR management tool that records the socio-ecological commitment of a company in a comparable way. Previous research indicates that companies with published CGBs provide elements of good working conditions [20]. The companies report about positive developments on the organizational level since publishing their first CGB [21,24]. Yet, these development reports are from exploratory studies and depict the perspective of a few persons from each company, which are mostly people with leadership roles. With our study, we enrich the research on ECG, because for the first time we captured the perspective of employees from ECG companies on a large scale and revealed the effects of a corporate common good-orientation on the micro level.

According to our results, employees in companies with higher CBG scores perceive more CSR. Additionally, correlations between the scores in the CGBs and work related attitudes and behaviors can be found. We were able to show that an increasing corporate common good-orientation in the sense of the ECG has a positive influence on employees' job satisfaction. Employees from companies with better job quality according to ECG standards feel better supported by their companies and experience less demands at work. A fair distribution of income according to the ECG criteria leads to higher satisfaction with wages. We observed that the value and social impact of the products and services of the company has an influence on how meaningful the work in the company is assessed, but not on the extent to which employees identify with their companies. However, with increasing corporate democracy and transparency, employees' organizational identification improves. Employees from companies that are behaving more co-operatively with other companies are more willing to take on tasks that are not part of their official job requirements but serve the functioning of the company. Yet, these employees do not behave more cooperatively with each other than employees from companies that behave less cooperatively with other companies. Overall, we interpret the results of our study as illustrating that an increasing corporate orientation towards the common good in the sense of the ECG can have a positive influence on employee's attitudes and behaviors.

We conclude that the CGB captures aspects that have an impact on job and pay level satisfaction, job demands, perceived organizational support, work meaningfulness, organizational citizenship behaviors towards the company and partly organizational identification. Even though the CGB ensures comparability in common good-orientation between the companies, interpretations of our results indicate that the CGB does not include relevant factors that determine job satisfaction in its scoring. Since employees are one of the most important stakeholders of a company, the CGB should not only be used to assess the objective working conditions in the company, but criteria that reflect the subjective well-being of employees should also be included in the CGB scoring in its further development.

**Author Contributions:** Conceptualization: J.W., K.H.; methodology: J.W.; formal analysis: J.W., K.H.; investigation: J.W.; data curation: J.W.; writing—original draft preparation: J.W.; writing—review and editing: K.H., supervision: K.H.; project administration, J.W. All authors have read and agreed to the published version of the manuscript.

**Funding:** The publication of this article was funded by Freie Universität Berlin.

**Institutional Review Board Statement:** Not applicable.

**Informed Consent Statement:** Not applicable.

**Data Availability Statement:** Not available.

**Acknowledgments:** We express our gratitude to the anonymous reviewers and also thank Paula Bleick who supported us in data collection, Andreas Stollberg and Patrick Krennmair for their statistical advice and Scapha Translations for translation work.

**Conflicts of Interest:** The authors declare no conflict of interest.

## Appendix A

**Table A1.** Example items used in the study.

| |
|---|
| Perceived CSR (from Glavas & Kelley, 2014; responses on a 5-level Likert scale from strongly disagree to strongly agree) |

Social dimension:
- Contributing to the well-being of employees is a high priority in my organization.
- Contributing to the well-being of suppliers is a high priority in my organization.

Ecological dimension:
- Environmental issues are integral to the strategy of my organization.
- My organization takes great care that our work does not hurt the environment

Job Satisfaction (from Judge & Klinger, 2008):
- All things considered, are you satisfied with your present job? No/yes
- How satisfied are you with your job in general? (Responses on a 5-level Likert scale from very dissatisfied to very satisfied)

Job Demands (from Fischer & Lück, 2014; responses on a 5-level Likert scale from very dissatisfied to very satisfied or from false to correct):

Often, too much is expected from us at work.

Are you satisfied with the pace of work? (inverse)

Pay Level Satisfaction (from Fischer & Lück, 2014; responses on a 5-level Likert scale from very dissatisfied to very satisfied):
- Are you satisfied with your pay?
- Are you satisfied with your pay when you compare it with that of your colleagues?

Perceived Organizational Support (from Eisenberger et al., 2001; responses on a 5-level Likert scale from strongly disagree to strongly agree):

- My company really cares about my well-being.
- My company strongly considers my goals and values.

Meaningful Work (from Steger et al., 2012 and Bunderson & Thompson, 2009; responses on a 5-level Likert scale from absolutely untrue to absolutely true).

Positive meaning:

- I have found a meaningful career.
- I have discovered work that has a satisfying purpose.

Meaning-making through work:

- I view my work as contributing to my personal growth.
- My work helps me better understand myself.

Greater good motivations:

- I know my work makes a positive difference in the world.
- The work I do serves a greater purpose.

Organizational Identification (from Mael & Ashfort, 1992; response on a 5-level Likert scale from strongly disagree to strongly agree):

- When someone criticizes my company, it feels like a personal insult.
- This company's successes are my successes.

Organizational Citizenship Behavior (OCB) (from Lee & Allen (2002); responses on a 5-level Likert scale from never to always)

OCB-I:

- I voluntarily take time to help others who have work-related problems.
- I support others with their tasks.

OCB-O:

- I take on tasks that are not required of me but that contribute to my company's image.
- I take measures to protect my company from potential problems.

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
