# Peer review of "The Common Good Balance Sheet and Employees’ Perceptions, Attitudes and Behaviors"

_sustainability, doi:10.3390/su13031592_

Round 1

Reviewer 1 Report

The article presents an interesting and important topic. The authors have a very clear insight into the theme and make a very interesting analysis from many points of view.

Although overall the article seems a piece of art from a sociological and psychological point of view, I still have serious concerns regarding the statistical analysis. It seems that data is "stretched' to prove the evidence.

First of all, I am not sure that the sample is representative. In table Table 2. N=332 is pretty high (it might be 350 for a big country like Germany), but the % of the total employee sin sample is very unbalanced. Also, the role of sex and even age in relation to CSR is not very clear. Even the authors explain that in section: 1.3.1. The Role of Sex and Age in Reactions to CSR. Even the authors try to show this confusing hypothesis in limitation: "Neither are we able to rule out gender bias, for two-thirds of the respondents, are female. According to the study by Ollé-Espluga et al. [20], women represent almost half of the workforce in ECG firms. However, according to the valid frequency in Ol060 lé-Espluga et al.’s [20] study, they account for two-thirds of the total workforce".

The second concern refers to all equation regressions, where R2 is extremely low. For example in section 3.1. although the model is significant R2 is 0.13a and Adj R2 is 0,11, meaning that only 11% of the variance of the dependent variable (PCSR =perceived corporate social responsibility) is explained by the variance of independent variables (CGBscore, age, and sex).
If I were the author, I would exclude sex from the model and maybe age, too, and check if the R2 would reach at least 0.6 value. Age and sex might have an influence, but with a low weight. In the new regression PCSR will depends only on CGBscore.

Another solution would be to delete all the regressions and keep the interpretation based especially on the correlation matrix (Table 3).
I was also amazed on behalf of the method applied to get so many answers, regarding so many criteria from employees. How did you motivate them? You may add some details in the methodology section.

Regarding references, there are too many very old references. The authors might cite papers published in WoS, Scopus, etc in the last 3 years.

Author Response

Dear reviewer,

Thank you very much for the effort you put into your review and your helpful comments and suggestions.

We would like to address your comments and suggestions:

Regarding the comments on the statistical analysis: We have carried out the analyses very carefully and diligently, making all steps and shortcomings of the study transparent. Although being firm in the usage of the methodology ourselves, we also asked two independent statisticians during analyses and regarding the reviews for their opinion. They supported our approach. With this, we are confident that from a statistical point of view the model and the analyses fit the available data and the data structure and analyses were carried out correctly.

Regarding the comments on variance explained: According to the mathematical definition of R2, removing variables will lower the R2. We ran an empirical check which confirms this: A calculation of the model of perceivedCSR = ScoreCGB + age + sex without age and sex leads to the result that R2 decreases (from .13 to .11). We agree, that the he explanatory power of the models is low, but it should be noted that this is empirical data from the field, and thus expectations on the explanatory power should should be adjusted. Age and gender have an influence on the experience of CSR. However, the directions of the effects have not yet been clearly elucidated. We have now clarified this fact in the paper. In addition, we already made two suggestions in the paper on how the explanatory power of the models may be improved in future studies.

Regarding the comments on the sample: It is true that more women than men participated in our survey (62%) and therefore we cannot exclude gender bias. However, the study by Ollé-Espluga et al. showed that the proportion of women in ECG companies is between 47% and 67%, so our study reflects the empirical gender ratio in ECG companies or comes very close to it. With these explanations, we hope to address your concern about representativeness. Even if the study is not representative, it still has explanatory power. We report both the b-values and the ß-values from the analyses, so that information is available for the present empirical data set as well as possible inferential comparisons.

Further comments and suggestions:

We added information in the methodology section on how we motivated companies to join our study and what might have motivated them to encourage their employees to fill out our survey.

We updated some old references. The remaining older references usually refer to the definitions of the constructs and theories studied, which are also used in this way in other current studies.

Changes in the paper are highlighted.

Thank you again for helping to improve the article. We hope that we could address all your concerns.

We wish you all the best for the new year.

Best regards,

the Authors

Reviewer 2 Report

Dear authors,
After a careful reading of your article I have the following considerations to make:

ABSTRACT:
I like to see the methodology used in the abstract introduced. The reader immediately realizes the type of analysis that was performed. I leave this suggestion to the authors.

INTRODUCTION:

The authors have chosen to join the introduction and the literature review (LR) where they have inserted the hypotheses. Personally I like to see the Introduction separated from LR. I think the article is more organized and easy to read.

I suggest that at the end a graphic image with the proposed research model be inserted.

A note to congratulate the authors for the excellent LR performed.

Translated with www.DeepL.com/Translator (free version)

METHODS

I ask the authors to make available the questionnaire used in appendix because it will make what was studied more perceptible and may also serve for future studies of other researchers, in other realities. The availability of the questions during the article makes the work very extensive and difficult to read. The article will be better organized with the availability of the survey questions at the end of the paper.

CONCLUSIONS

The article presents results and a very complete discussion. Perhaps even too extensive and somewhat exaggerated in explanations.
Then, surprisingly, it presents a poor conclusion.
Why? Did the authors get tired of writing? I think there must be a balance between the sections. For this reason I suggest that the conclusions be completed because the article deserves to end with a robust and scientifically comprehensive conclusion. They should approach the research contributions in a coherent way.

In short: I liked your article very much. I hope to be able to accept it in the next round of review.

Good luck in your research and academic career.

Wishes for an excellent 2021

Author Response

Dear reviewer,

Thank you very much for the effort you put into your review and your helpful comments and suggestions.

We would like to address your comments and suggestions:

We added information about the methodology used in the abstract.

We separated the introduction from the literature review and following paragraphs to improve the article´s structure.

We inserted a graphic image with the proposed research model.

We agree with you that it is easier to read if the items are not mentioned in the methods section but are separate in an appendix. Showing all the items we have used will be problematic from a copyright point of view. We have included sample items, so that the readers get an idea of what the scale contains. The reference in the methods section will then lead them to the full scale in case they are interested.

We have updated and refined the conclusion and hope that is clearer and more comprehensive now.

Changes in the paper are highlighted.

Thank you again for helping to improve the article. We hope that we could address all your concerns.

We wish you all the best for the new year.

Best regards,

the Authors